# Fitness Cost of the Field-Evolved Resistance to Sulfoxaflor and Multi-Insecticide Resistance of the Wheat Aphid *Sitobion miscanthi* (Takahashi)

**DOI:** 10.3390/insects14010075

**Published:** 2023-01-12

**Authors:** Xinan Li, Saige Zhu, Qiuchi Li, Yulin Sun, Yanbo Wang, Xujun Tian, Xiao Ran, Xiangrui Li, Yunhui Zhang, Haifeng Gao, Xun Zhu

**Affiliations:** 1State Key Laboratory for Biology of Plant Diseases and Insect Pests, Institute of Plant Protection, Chinese Academy of Agricultural Sciences, Beijing 100193, China; 2Henan Engineering Research Center of Green Pesticide Creation & Intelligent Pesticide Residue Sensor Detection, School of Resource and Environmental Sciences, Henan Institute of Science and Technology, Xinxiang 453003, China; 3Key Laboratory of Integrated Pest Management on Crop in Northwestern Oasis, Ministry of Agriculture and Rural Affairs, Institute of Plant Protection, Xinjiang Academy of Agricultural Sciences, Urumqi 830091, China

**Keywords:** fecundity, fitness cost, insecticide resistance, *Sitobion miscanthi*, sulfoxaflor

## Abstract

**Simple Summary:**

Pest control mainly relies on the use of chemical insecticides, whereas insecticide resistance is problematic for the effective management of insect pests. Sulfoxaflor is a new sulfoximine insecticide. The wheat aphid *Sitobion miscanthi* is one of the most important pests adversely affecting wheat cultivation. We obtained an *S*. *miscanthi* field population highly resistant to sulfoxaflor, despite this insecticide not having been widely and continuously used for control of wheat aphids in China. The understanding of the cross-resistance or multi-resistance spectrum and fitness cost caused by insecticide resistance is important for selecting suitable insecticides to integratively manage insecticide resistance. Our findings suggest that the *S*. *miscanthi* population that is highly resistant to sulfoxaflor had moderate resistance to two pyrethroid insecticides, which was accompanied by severely adverse biological fitness. The study provides valuable information regarding the rational use of pesticides and may be relevant for exploring new mechanisms of insecticide resistance.

**Abstract:**

Sulfoxaflor belongs to a new class of insecticides that is effective against many sap-feeding pests. In this study on *Sitobion miscanthi* (Takahashi) (i.e., the predominant wheat pest), a highly sulfoxaflor-resistant (SulR) population was obtained from a field. Its resistance to the other seven insecticides and its biological fitness were analyzed using a leaf-dip method and a two-sex life table approach, respectively. Compared with the relatively susceptible (SS) population, the SulR population was highly resistant to sulfoxaflor, with a relative insecticide resistance ratio (RR) of 199.8 and was moderately resistant to beta-cypermethrin (RR = 14.5) and bifenthrin (RR = 42.1) but exhibited low resistance to chlorpyrifos (RR = 5.7). Additionally, the SulR population had a relative fitness of 0.73, with a significantly prolonged developmental period as well as a lower survival rate and poorer reproductive performance than the SS population. In conclusion, our results suggest that *S*. *miscanthi* populations that are highly resistant to sulfoxaflor exist in the field. The possibility that insects may develop multi-resistance between sulfoxaflor and pyrethroids is a concern. Furthermore, the high sulfoxaflor resistance of *S*. *miscanthi* was accompanied by a considerable fitness cost. The study data may be useful for improving the rational use of insecticides and for exploring novel insecticide resistance mechanisms.

## 1. Introduction

Sulfoxaflor is a new sulfoximine insecticide that is highly effective against a variety of sap-feeding insect pests [1]. It affects the nicotinic acetylcholine receptors in the insect nervous system, but its binding site differs from that of other insecticides, including neonicotinoids, spinosyns, and nereistoxin analogs [2]. Additionally, sulfoxaflor was designated as the only Group 4C new active ingredient by the Insecticide Resistance Action Committee [3]. Moreover, sulfoxaflor is a nicotinic acetylcholine receptor agonist that induces hyperactivity and paralysis in insects, resulting in death [4]. It can be used to control various insect pests (e.g., Aphidoidea, Miridae, Thripoidea, and Delphacidae species) that infest wheat, cotton, rice, fruit trees, and other crops, while also controlling pests resistant to neonicotinoid, organophosphate, pyrethroid, and carbamate insecticides [2].

Pest control measures typically involve the widespread use of chemical insecticides, which often leads to insect populations developing resistance to those insecticides. Earlier research revealed that sulfoxaflor performs equally well against *Nilaparvata lugens* and *Bemisia tabaci* populations susceptible and resistant to imidacloprid in the laboratory [1], but different field populations of *N*. *lugens* and *Aphis gossypii* have been identified with low resistance to sulfoxaflor [5,6]. In addition, a *B*. *tabaci* strain resistant to pyrethroid (deltamethrin) and organophosphate (prophosphate) insecticides has not shown cross-resistance to sulfoxaflor, and there is no cross-resistance to sulfoxaflor and neonicotinoids (imidacloprid) in *Trialeurodes vaporariorum* [7]. However, an *N*. *lugens* strain resistant to sulfoxaflor in the laboratory reportedly exhibited considerable cross-resistance to dinotefuran, nitenpyram, thiamethoxam, clothianidin, imidacloprid, and cycloxaprid, but low or no cross-resistance to isoprocarb, etofenprox, chlorpyrifos, triflumezopyrim, and buprofezin [8].

The insecticide resistance of insect populations is usually accompanied by a fitness cost (e.g., low survival rate, fecundity, hatching, and longevity), which may affect how quickly insect populations develop resistance [9,10,11]. This fitness cost is considered to reflect the evolution of insect resistance. Thus, thoroughly characterizing this fitness cost is critical for developing effective insecticide resistance management strategies [12,13,14]. Life table analyses have been widely used in ecological studies, including those examining the timing of pest control procedures [15], host preferences, and insect fitness [16]. Previous studies have examined the fitness costs of *Myzus persicae*, *N*. *lugens*, and *A*. *gossypii* strains resistant to sulfoxaflor [6,8,17].

The wheat aphid *Sitobion miscanthi* (Fabricius) (Hemiptera: Aphididae) is one of the most important pests adversely affecting wheat cultivation in China [18,19,20]. To the best of our knowledge, sulfoxaflor has not been widely and consistently used for the long-term control of *S*. *miscanthi* in China [21]. However, during an investigation of the resistance levels of *S*. *miscanthi* field populations to insecticides throughout China, we detected *S*. *miscanthi* field populations that were highly resistant to sulfoxaflor [22]. In this study, the fitness cost of field-evolved resistance to sulfoxaflor and the resistance to multiple insecticides were investigated using *S*. *miscanthi* populations. The study provides valuable information regarding the rational use of pesticides and may be relevant for exploring new mechanisms of insecticide resistance.

## 2. Materials and Methods

### 2.1. Insects and Insecticides

The field population of *S*. *miscanthi* with high resistance to sulfoxaflor (SulR) was collected from Kunming, Yunnan province, China, in 2019 (N24°59′58″, E102°33′11″). The sulfoxaflor-susceptible (SS) field population of *S*. *miscanthi* was collected from Hefei, Anhui province, China, in 2019 (N31°57′34″, E117°11′36″). All populations were reared on wheat seedlings (Lunxuan 987) in a climate-controlled chamber maintained at 20 ± 1 °C with 60 ± 10% relative humidity and a 16 h light:8 h dark photoperiod. The aphid populations were not exposed to any pesticides.

Sulfoxaflor (96%) was supplied by Hubei Kangbaotai Fine Chemicals Co., Ltd. (Wuhan, China), whereas imidacloprid (96%), beta-cypermethrin (95%), thiamethoxam (97%), bifenthrin (97%), abamectin (95%), and chlorpyrifos (97%) were provided by Beijing Green Agricultural Science and Technology Group Co., Ltd. (Beijing, China). Omethoate (40%, emulsifiable concentrate) was provided by Hebei Xinxing Chemical Co., Ltd. (Baoding, Hebei, China).

### 2.2. Insecticide Bioassays

The toxicity of sulfoxaflor, neonicotinoids (imidacloprid and thiamethoxam), pyrethroids (beta-cypermethrin and bifenthrin), organophosphates (chlorpyrifos and omethoate), and abamectin to field *S*. *miscanthi* populations was determined by performing insecticide bioassays using a leaf-dip method [23]. The insecticides were formulated as stock solutions (10,000 mg/L) with acetone and then stored at 4 °C. Each insecticide was diluted with 0.1% Tween-80 solution (prepared in water) to produce working solutions (five concentrations), with three replicates per concentration. Controls were treated with the 0.1% Tween-80 solution. For each insecticide concentration, at least 30 aphids were treated. Specifically, the apterous aphids together with the leaves were immersed in the insecticide working solutions for 3–5 s, after which they were placed on moistened filter paper in a disposable culture dish. Aphids were incubated at 20 ± 1 °C with 60 ± 10% relative humidity and a 16 h light:8 h dark photoperiod. Aphid mortality was examined using a stereomicroscope at 24 h after the treatment. Aphids were considered dead if they were unable to move after being touched with an anatomical needle.

### 2.3. Fitness Comparison

Life tables for the SulR and SS *S*. *miscanthi* populations were established using the age-stage, two-sex life table approach [24]. For each population, 120 apterous adults that had not reproduced were placed on fresh wheat seedlings in a Petri dish (9 cm diameter) lined with moistened filter paper to facilitate reproduction, with one adult per dish. After 24 h, all adult aphids were removed, and a random nymph was left in each dish. Population parameters, including developmental time, fecundity, mortality, and longevity were analyzed daily. During the reproductive period, the number of newborn nymphs produced by females was recorded daily. The wheat seedlings were replaced by fresh seedlings every 3 days until all adult aphids died. The life table experiment was performed in a greenhouse at 20 ± 1 °C with 60 ± 10% relative humidity and a 16 h light:8 h dark photoperiod.

### 2.4. Data Analysis

The mortality data were adjusted based on the control mortality (<10%) using Abbott’s formula. The median lethal concentrations (LC_50_), 95% confidence intervals, and slopes were calculated using Data Processing System software (version 7.05) (Zhejiang University, Hangzhou, China). The relative insecticide resistance ratio (RR) of the two *S*. *miscanthi* populations was calculated by dividing the LC_50_ of the SulR population by the LC_50_ of the SS population. Insecticide resistance, including multi-resistance, was classified according to the RR as follows: RR ≤ 5 (susceptible), 5 < RR ≤ 10 (low resistance), 10 < RR ≤ 100 (moderate resistance), and RR > 100 (high resistance) [25,26].

The life table data of two *S*. *miscanthi* populations were analyzed according to age-stage, two-sex life table theory [24,27,28]. The developmental duration, longevity, reproductive days, fecundity, age-stage-specific survival rate (*s_xj_*), age-specific survival rate (*l_x_*), age-specific fecundity (*m_x_*), age-specific maternity (*l_x_m_x_*), age-specific life expectancy (*e_xj_*), age-stage reproductive value (*v_xj_*), intrinsic rate of increase (*r*), finite rate of increase (*λ*), net reproductive rate (*R*_0_), and mean generation time (*T*) were calculated using the TWOSEX-MS Chart program [27,28,29]. The variance and standard error of the life table parameters were calculated using the bootstrap procedure included in the TWOSEX-MS Chart, with 100,000 random resamplings [30,31,32,33]. The relative fitness (*R*_f_) of the SulR population was calculated as follows: *R*_f_ = *R*_0_ of SulR/*R*_0_ of SS [9,12]. An *R*_f_ value < 1 suggests that the resistance of the population was accompanied by a fitness cost [34].

## 3. Results

### 3.1. Field-Evolved Resistance of SulR and SS S. miscanthi Populations to Insecticides

The resistance of *S*. *miscanthi* field populations to sulfoxaflor was evaluated in a bioassay. The bioassay data were used to calculate the LC_50_ for the two populations. The RR indicated that compared with the SS population, the SulR population was highly resistant to sulfoxaflor (RR > 100) (Table 1). The resistance of the SulR *S*. *miscanthi* population to the following seven insecticides was assessed: neonicotinoids (imidacloprid and thiamethoxam), pyrethroids (beta-cypermethrin and bifenthrin), organophosphates (chlorpyrifos and omethoate), and abamectin. The SulR population exhibited moderate resistance to beta-cypermethrin and bifenthrin and low resistance to chlorpyrifos (Table 1). Interestingly, there was susceptibility to abamectin, imidacloprid, thiamethoxam, or omethoate (RR < 5) (Table 1).

### 3.2. Developmental Duration and Fecundity of SulR and SS S. miscanthi Populations

The developmental duration, longevity, and fecundity of the SulR and SS populations are presented in Table 2. There were no significant differences between the SulR and SS *S*. *miscanthi* populations regarding the developmental times, including the first, second, and third nymph stages (L1, L2, and L3, respectively), the reproductive period, adult longevity, or total longevity (*p* > 0.05). However, the fourth nymph stage (L4), the whole nymph stage (pre-adult), adult pre-reproductive period (APRP), and total pre-reproductive period (TPRP) were 0.36, 0.49, 0.19, and 0.57 days longer for the SulR population than for the SS population, respectively (*p* < 0.05). Moreover, on average, the SulR population produced 4.18 fewer offspring per aphid than the SS population (*p* < 0.05).

### 3.3. Comparison of s_xj,_ l_x_m_x,_ e_xj_, and v_xj_ between the SulR and SS S. miscanthi Populations

The *s_xj_* data for the SulR and SS populations are provided in Figure 1. The differences in the developmental duration between individuals lead to overlapping stages of development. The *s_xj_* of the third nymph stage (L3) was higher for the SulR population than for the SS population, but the opposite pattern was observed for the adult stage (female). The *m_x_* data indicated that both *S*. *miscanthi* populations began to produce offspring after day 5, with peak offspring production on day 9 and no offspring production after day 25 (Figure 2). The *l_x_*, *m_x_*, and *l_x_m_x_* values were lower for the SulR population than for the SS population from day 9 to 23, day 6 to 13, and day 6 to 13, respectively (Figure 2), indicating that the survival rate was lower for the SulR population than for the SS population during these periods. Additionally, the *e_xj_* and *v_xj_* data were lower for the SulR population than for the SS population (Figure 3 and Figure 4).

### 3.4. Population Life Table Parameters of the SulR and SS S. miscanthi Populations

The life table parameters of the SulR and SS populations are listed in Table 3. There were no significant differences between the SulR and SS populations regarding *T*, whereas *R*_0_, *r*, and *λ* were significantly lower for the SulR population than for the SS population (*p* < 0.05). Moreover, the *R*_f_ of the SulR population (0.73) indicated that the high sulfoxaflor resistance of this population adversely affected fitness.

## 4. Discussion

Insecticide-resistant insect strains have been isolated in many studies by the successive selection with insecticides under laboratory conditions. For example, sulfoxaflor-resistant strains of *A*. *gossypii* [6], *M*. *persicae* [17], and *N*. *lugens* [8] have been isolated. However, the importance of specific alleles for the development of insecticide resistance can only be confirmed in insect populations that evolved their resistance under natural conditions [35]. Therefore, insecticide-resistant field populations are preferable for studies because of their resistance mechanisms that likely evolved under field conditions [36]. Earlier investigations of field-evolved insecticide resistance involved, for example, *Ostrinia nubilalis* resistant to the Cry1Ab toxin [37]; *Helicoverpa armigera* resistant to the Cry1Ac toxin [38]; *Plutella xylostella* resistant to avermectin, spinosad, *Bacillus thuringiensis*, and chlorantraniliprole [39,40,41,42]; and *N*. *lugens* resistant to imidacloprid and ethiprole [43]. In the current study, a *S*. *miscanthi* population that was highly resistant to sulfoxaflor was obtained from a field.

Sulfoxaflor is a new sulfoximine insecticide that is highly effective against a variety of sap-feeding insect pests and has not shown cross-resistance with other existing insecticides [1,2,7]. Therefore, it is widely used in the integrated management of pests’ resistance to insecticides [1]. However, we observed that *S*. *miscanthi* field populations may become highly resistant to sulfoxaflor. Additionally, previous studies have shown that field populations of *N*. *lugens* and *A*. *gossypii* have developed low-level resistance to sulfoxaflor in different regions of China [5,6]. These findings indicate a major challenge for the effective use of sulfoxaflor in the control of insect pests.

An insect population that has developed resistance to one insecticide may develop resistance to other insecticides. In this study, the SulR *S*. *miscanthi* population exhibited moderate resistance to beta-cypermethrin and bifenthrin as well as low resistance to chlorpyrifos, but it was not resistant to abamectin, imidacloprid, thiamethoxam, or omethoate (Table 1). In a previous study, a sulfoxaflor-resistant *N*. *lugens* strain exhibited a high level of cross-resistance to thiamethoxam and moderate cross-resistance to imidacloprid, but no cross-resistance to chlorpyrifos [5]. In another study, a *B*. *tabaci* strain resistant to pyrethroid (deltamethrin) and organophosphate (prophosphate) insecticides was not cross-resistant to sulfoxaflor, and *T*. *vaporariorum* was not cross-resistant to sulfoxaflor and neonicotinoid (imidacloprid) insecticides [7]. These results indicate that the resistance to sulfoxaflor and other insecticides is inconsistent among insect species.

It is unlikely that the resistance of *S*. *miscanthi* to sulfoxaflor was the result of long-term insecticide applications because, to the best of our knowledge, sulfoxaflor has not been widely and consistently used for the long-term control of wheat aphids in the sample collection region [21]. In other words, the resistance of the SulR population to beta-cypermethrin and bifenthrin may be unrelated to its high level of resistance to sulfoxaflor. On the contrary, the pyrethroids are widely and continuously used for control of wheat aphids in China [21], and the resistance of field *S*. *miscanthi* population to sulfoxaflor are likely to be the result of selection by pyrethroids. Overall, our findings suggest that the SulR *S*. *miscanthi* population did not become highly resistant to other insecticides in the field. We strongly recommend that multiple rotations of sulfoxaflor and pyrethroids should be avoided when sulfoxaflor is used to control *S*. *miscanthi*.

The development of insecticide resistance is often accompanied by changes in life table parameters. In previous studies, the developmental and the pre-adult durations of sulfoxaflor-resistant *A*. *gossypii* and *M*. *persicae* strains were shorter than the corresponding periods in susceptible strains [6,17]. However, these findings are inconsistent with those of an investigation of the developmental duration of sulfoxaflor-resistant *N*. *lugens* strains [44]. In the current study, the L4 and pre-adult stages as well as the APRP and TPRP were significantly longer for the SulR population than for the SS population (*p* < 0.05) (Table 2), which is consistent with the developmental duration of the sulfoxaflor-resistant *N*. *lugens* strains. Although the reproductive period, adult longevity, and total longevity did not significantly differ between the SulR and SS populations, the fecundity of the SulR population was significantly lower than that of the SS population (Table 2).

In this study, the *s_xj_*, *l_x_*, *m_x_*, *l_x_m_x_*, *e_xj_*, and *v_xj_* values were lower for the SulR population than for the SS population (Figure 1, Figure 2, Figure 3 and Figure 4), suggesting that the development of sulfoxaflor resistance decreased the survival and reproduction of *S*. *miscanthi*. This is in accordance with the findings of most previous related studies, which revealed that the development of insecticide resistance is associated with significant disadvantages [9,45]. Therefore, the sulfoxaflor resistance of the SulR *S*. *miscanthi* population adversely affected fitness, likely because of a trade-off in terms of resource allocation. Although it needs to be confirmed in further studies, the cross-resistance and fitness reduction indicates metabolic resistance as more likely to be the type of resistance in this population.

Furthermore, *R*_0_, *T*, *r*, and *λ* are important indices for assessing the biological characteristics of insect populations [9]. These indices appear to indicate that insecticide resistance strongly influences life history traits, which helps to delay the evolution of additional insecticide resistance [9]. In the current study, *T* did not differ substantially between the SulR and SS populations, whereas there were significant differences between the two populations regarding *R_0_*, *r*, and *λ* (*p* < 0.05). Moreover, the *R*_f_ of the SulR population was 0.73 (Table 3). Therefore, the cost of the field-evolved sulfoxaflor resistance of the SulR *S*. *miscanthi* population was a substantial decrease in fitness.

In conclusion, our findings indicate that the *S*. *miscanthi* population that is highly resistant to sulfoxaflor exhibited moderate resistance to beta-cypermethrin and bifenthrin but exhibited low resistance to chlorpyrifos. In addition, insecticide resistance adversely affected biological fitness. The extensive use and misuse of chemical insecticides may be responsible for the rapid evolution of high-level resistance among the insects in the sample collection region [39]. However, the *S*. *miscanthi* field populations developed high levels of resistance to sulfoxaflor even though this insecticide has not been widely and continuously used over a prolonged period. This may reflect the diverse ways in which insects develop insecticide resistance. The field-evolved resistance of *S. miscanthi* to sulfoxaflor and the associated mechanisms will need to be more thoroughly investigated.

## Figures and Tables

**Figure 1 insects-14-00075-f001:**
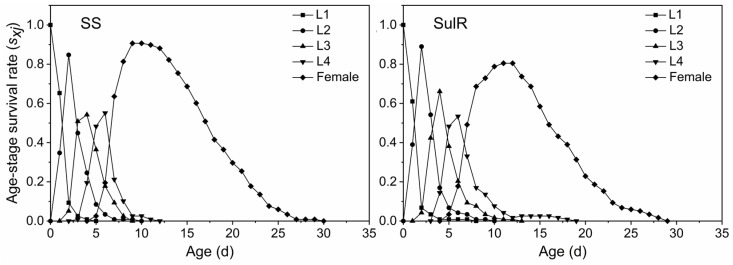
Age-stage-specific survival rate (*s_xj_*) of sulfoxaflor-resistant (SulR) and sulfoxaflor-susceptible (SS) *Sitobion miscanthi* populations. L1, first nymph stage; L2, second nymph stage; L3, third nymph stage; L4, fourth nymph stage.

**Figure 2 insects-14-00075-f002:**
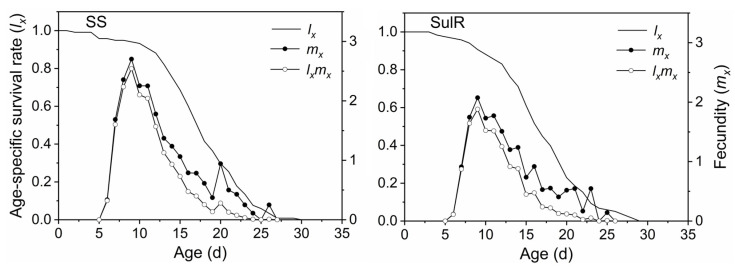
Age-specific survival rate (*l_x_*), age-specific fecundity of the total population (*m_x_*), and age-specific maternity (*l_x_m_x_*) of sulfoxaflor-resistant (SulR) and sulfoxaflor-susceptible (SS) *Sitobion miscanthi* populations. L1, first nymph stage; L2, second nymph stage; L3, third nymph stage; L4, fourth nymph stage.

**Figure 3 insects-14-00075-f003:**
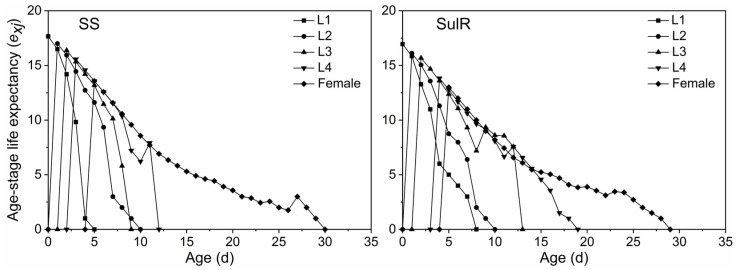
Age-stage-specific life expectancy (*e_xj_*) of sulfoxaflor-resistant (SulR) and sulfoxaflor-susceptible (SS) *Sitobion miscanthi* populations. L1, first nymph stage; L2, second nymph stage; L3, third nymph stage; L4, fourth nymph stage.

**Figure 4 insects-14-00075-f004:**
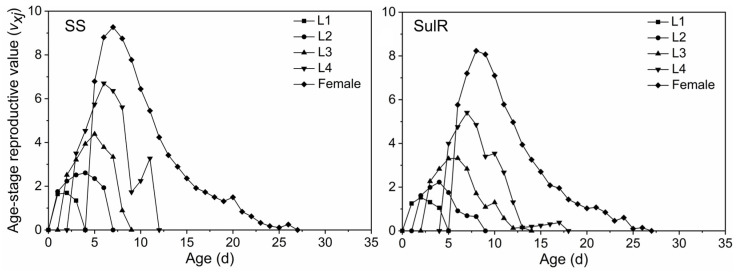
Age-stage-specific reproductive rate (*v_xj_*) of sulfoxaflor-resistant (SulR) and sulfoxaflor-susceptible (SS) *Sitobion miscanthi* populations. L1, first nymph stage; L2, second nymph stage; L3, third nymph stage; L4, fourth nymph stage.

**Table 1 insects-14-00075-t001:** Toxicity of the tested insecticides to *Sitobion miscanthi* field populations.

Insecticide	Population	*N* ^a^	Slope ± SE ^b^	LC_50_ ^c^ (95% CI ^d^; mg/L)	*χ* ^2^	*p*-Value	RR ^e^
Sulfoxaflor	SS	708	0.51 ± 0.11	1.7 (0.2–4.6)	0.26	0.88	
	SulR	585	0.53 ± 0.10	339.6 (138.1–1835.7)	2.30	0.32	199.8
Imidacloprid	SS	542	0.40 ± 0.06	37.4 (16.0–82.5)	0.34	0.84	
	SulR	598	0.42 ± 0.07	70.3 (33.62–184.7)	3.10	0.21	1.9
Thiamethoxam	SS	614	0.56 ± 0.06	23.3 (13.6–37.4)	6.80	0.08	
	SulR	567	0.67 ± 0.08	37.8 (23.8–62.9)	0.79	0.85	1.6
Beta-cypermethrin	SS	605	0.63 ± 0.06	4.8 (2.5–8.0)	6.40	0.09	
	SulR	541	0.77 ± 0.07	69.5 (45.9–112.9)	1.00	0.80	14.5
Bifenthrin	SS	587	0.66 ± 0.07	3.5 (1.7–6.0)	5.54	0.14	
	SulR	584	0.51 ± 0.07	147.4 (73.8–404.6)	0.69	0.88	42.1
Abamectin	SS	623	0.86 ± 0.10	12.2 (8.5–17.8)	1.72	0.42	
	SulR	645	1.31 ± 0.15	9.8 (7.0–14.0)	0.56	0.76	0.8
Chlorpyrifos	SS	595	2.28 ± 0.22	2.3 (1.9–2.8)	5.29	0.15	
	SulR	615	2.76 ± 0.42	13.2 (11.6–14.9)	0.37	0.54	5.7
Omethoate	SS	593	1.24 ± 0.11	97.5 (7.05–140.8)	6.09	0.11	
	SulR	655	1.46 ± 0.15	263.4 (192.4–386.9)	0.30	0.86	2.7

^a^ number of tested aphids; ^b^ standard error; ^c^ concentrations (mg/L) resulting in 50% dead or affected insects after 24 h; ^d^ 95% confidence limit of the median lethal concentrations; ^e^ relative insecticide resistance ratio, RR ≤ 5 (susceptible), 5 < RR ≤ 10 (low resistance), 10 < RR ≤ 100 (moderate resistance), and RR > 100 (high resistance).

**Table 2 insects-14-00075-t002:** Developmental duration and fecundity of sulfoxaflor-susceptible (SS) and sulfoxaflor-resistant (SulR) *Sitobion miscanthi* populations.

Parameter ^a^	SS Population		SulR Population		*p*-Value
	N ^b^	Mean ± SE	N	Mean ± SE	
L1 (d)	120	1.75 ± 0.06 a	120	1.75 ± 0.08 a	0.9705
L2 (d)	113	2.01 ± 0.09 a	112	2.08 ± 0.08 a	0.5565
L3 (d)	110	1.84 ± 0.07 a	109	2.02 ± 0.08 a	0.0940
L4 (d)	109	1.72 ± 0.07 b	103	2.08 ± 0.07 a	0.0005
Pre-adult (d)	109	7.24 ± 0.10 b	103	7.73 ± 0.18 a	0.0173
Adult longevity (d)	109	11.34 ± 0.42 a	103	10.34 ± 0.46 a	0.1075
Total longevity (d)	118	17.66 ± 0.48 a	118	16.93 ± 0.48 a	0.2851
APRP (d)	107	0.45 ± 0.06 b	97	0.64 ± 0.06 a	0.0328
TPRP (d)	107	7.63 ± 0.11 b	97	8.20 ± 0.17 a	0.0055
Reproductive period (d)	109	7.50 ± 0.32 a	103	6.74 ± 0.31 a	0.0889
Fecundity (offspring/female)	109	18.40 ± 0.98 a	103	14.22 ± 1.00 b	0.0031

^a^ L1, first nymph stage; L2, second nymph stage; L3, third nymph stage; L4, fourth nymph stage; Pre-adult, complete nymph stage; APRP, adult pre-reproductive period; TPRP, total pre-reproductive period. ^b^ number of tested aphids. Data are presented as the mean ± SE. Values in the same row followed by different letters are significantly different (*p* < 0.05) from the respective control values.

**Table 3 insects-14-00075-t003:** Life table parameters of sulfoxaflor-susceptible (SS) and sulfoxaflor-resistant (SulR) *Sitobion miscanthi* populations.

Parameter	SS Population		SulR Population		*p*-Value
	n	Mean ± SE	n	Mean ± SE	
Net reproductive rate (*R*_0_)	120	17.00 ± 1.00 a	120	12.41 ± 0.98 b	0.0011
Mean generation time (*T*)	120	10.93 ± 0.14 a	120	11.28 ± 0.14 a	0.0790
Intrinsic rate of increase (*r*)	120	0.26 ± 0.01 a	120	0.22 ± 0.01 b	0.0003
Finite rate of increase (*λ*)	120	1.30 ± 0.01 a	120	1.25 ± 0.01 b	0.0003
Relative fitness (*R*_f_) ^a^				0.73	

^a^*R*_f_ = *R*_0_ of the SulR population/*R*_0_ of the SS population. Data are presented as the mean ± SE. Values in the same row followed by different letters are significantly different (*p* < 0.05) from the respective control values.

## Data Availability

The data presented in this study are available in article.

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
