# Peer review of "Fitness Cost of the Field-Evolved Resistance to Sulfoxaflor and Multi-Insecticide Resistance of the Wheat Aphid Sitobion miscanthi (Takahashi)"

_insects, 2023, doi:10.3390/insects14010075_

Round 1

Reviewer 1 Report

The authors found a field population of wheat aphid showed resistance to multiple insecticides especially sulfoxaflor, and the population also demonstrated significant fitness cost such as prolonged developmental period, lower survival rate and poorer reproduction. The results are helpful for scientific selection and application of insecticides in wheat aphid control. The manuscript is acceptable after a minor revision.

1.     Table 1: please confirm the right P values were used. A P value <0.05 means the DP-Line was no passed the Chi-square test and the bioassay must be re-done.

Author Response

1. Table 1: please confirm the right P values were used. A P value <0.05 means the DP-Line was no passed the Chi-square test and the bioassay must be re-done.

Author’s Response: The data in Table 1 were obtained using the variable data machine value analysis method. Accordingly, the P-value is less than 0.05. We used an attribute data analysis method to re-analyze the data using the DPS software (version 7.05) and revised the P-values in Tables 1.

Reviewer 2 Report

The manuscript provides relevant information that can contribute to the understanding and management of resistance.

Line 23: How long is a long period?, Please add this information.

Line 106: It is importaant to mention the generation of the field strain used for the bioassays.

Line 225: It would be interesting if they included why resistance was presented to an OP and not to another of the tested ones.

Author Response

Line 23: How long is a long period?  Please add this information.

Author’s Response: We have revised in the text. 

line 23-24: “in spite of this insecticide has not been used widely and continuously for control of wheat aphids in China”

Line 106: It is important to mention the generation of the field strain used for the bioassays.

Author’s Response: We have revised in the text. 

Line 118: “to field S. miscanthi populations was determined”

Line 225: It would be interesting if they included why resistance was presented to an OP and not to another of the tested ones.

Author’s Response: We will continue to carry out relevant research in the next step. 

Reviewer 3 Report

The manuscript reported a study on investigating the fitness cost of the field-evolved resistance to sulfoxaflor and multi-insecticide resistance in the wheat aphid Sitobion miscanthi (Takahashi). The present study is overall well presented and sound in its content. Their findings will provide valuable information for the effective management of S. miscanth resistant to insecticides including sulfoxaflor. However, this manuscript has some concerns which needed to be addressed as follows:

Major concerns: 

1. In Lines 101-105, the authors should provide the information of abamectin. Additionally, they should explain why they selected a commercial formulation (40% EC) of omethoate instead of its technical material (TC) because TC was used for other seven insecticides ?

2. In 138-141, the resistance levels of an insecticide tested should be classified in detail, at least including susceptibility and even deceased susceptibility. In addition, I strongly recommended the authors provided the corresponding reference.

Minor concerns:

1. In Lines 25-26, to integratively manage insecticide resistance” should be better than and integrated management of insecticide resistance.

2. In Line 27, accompany with should be revised as accompanies with.

3. In Lines 28-29 and 89-90, The study generated valuable information regarding the rational use of pesticides should be revised as The study provides valuable information for regarding the rational use of pesticides.

4. In Line 32, seven insecticides should be revised as other seven insecticides or eight insecticides.

5. In Lines 40-41, The possibility insects may develop multi-resistance between to sulfoxaflor and pyrethroids is a concern. should be revised as The possibility that insects may develop multi-resistance to sulfoxaflor and pyrethroids is a concern..

6. In Line 58, also controlling should be revised as to also control.

In Line 65 and 109, both  B. tabaci and S. miscanthi should be in italic.

7. In Line 66, has maybe better than is reportedly.

8. In Line 95, a field in should be deleted.

9. In Table 1, the first lower-case letter of sulfoxaflor should be replaced by an upper-case letter.

Finally, I hope the authors should use these to correct the same problem for the rest.

Author Response

Major concerns:

1. In Lines 101-105, the authors should provide the information of abamectin. Additionally, they should explain why they selected a commercial formulation (40% EC) of omethoate instead of its technical material (TC) because TC was used for other seven insecticides?

Author’s Response:

(1) We have revised in the text.  line 111: “abamectin (95%)”

(2) Because it was difficult to get technical material (TC) of omethoate at that time, we used commercial formulation (40% EC) of omethoate.

2. In 138-141, the resistance levels of an insecticide tested should be classified in detail, at least including susceptibility and even deceased susceptibility. In addition, I strongly recommended the authors provided the corresponding reference.

Author’s Response:  We have revised in the text and provided the corresponding reference. 

line 148-150: “RLR ≤ 5 (susceptible), 5 < RLR ≤ 10 (low resistance), 10 < RLR ≤ 100 (moderate resistance), and RLR > 100 (high resistance) [25,26]”

Minor concerns:

1. In Lines 25-26, “to integratively manage insecticide resistance” should be better than “and integrated management of insecticide resistance”.

Author’s Response:  We have revised in the text.

Line 26: “to integratively manage insecticide resistance”

2. In Line 27, “accompany with” should be revised as “accompanies with”.

Author’s Response:  We have revised in the text.

Line 28: “accompanies with”

3. In Lines 28-29 and 89-90, “The study generated valuable information regarding the rational use of pesticides” should be revised as “The study provides valuable information for regarding the rational use of pesticides”.

Author’s Response:  We have revised in the text.

Line 29-30, 92-93: “The study provides valuable information for regarding the rational use of pesticides”

4. In Line 32, “seven insecticides” should be revised as “other seven insecticides” or “eight insecticides”.

Author’s Response:  We have revised in the text.

Line 34: “Its resistance to other seven insecticides”

5. In Lines 40-41, “The possibility insects may develop multi-resistance between to sulfoxaflor and pyrethroids is a concern.” should be revised as “The possibility that insects may develop multi-resistance to sulfoxaflor and pyrethroids is a concern.”.

Author’s Response:  We have revised in the text.

Line 42: “The possibility that insects may develop multi-resistance to sulfoxaflor and pyrethroids is a concern.”

6. In Line 58, “also controlling” should be revised as “to also control”.

In Line 65 and 109, both “ B. tabaci” and “S. miscanthi” should be in italic.

Author’s Response:  We have revised in the text.

Line 60: “to also control”

In Line 67 and 118, “B. tabaci”. “S. miscanthi

7. In Line 66, “has” maybe better than “is reportedly”.

Author’s Response:  We have revised in the text.

Line 68: “has”

8. In Line 95, “a field in” should be deleted.

Author’s Response:  We have revised in the text. Line 97-100

9. In Table 1, the first lower-case letter of “sulfoxaflor” should be replaced by an upper-case letter.

Author’s Response:  We have revised in the Table 1. “Sulfoxaflor”

Reviewer 4 Report

Dear authors, 

I have completed reviewing your ms. You did a good job on this ms, making the reader able to follow the text easily, with conciseness, and presenting/discussing the results clearly. I don't see any concern on the data itself. I am also glad you did not oversell or inflate any conclusions, as your conclusions were draw well within limits of what your data presents. 

There are however few minor editing on the text, and a small insertion on discussion that I suggest, as presented in the attached pdf.

Author Response

1. Line 22, “an” should be revised as “a”.

Author’s Response:  We have revised in the text.

Line 22: “a”

2. Line 23, delete “has”

Author’s Response:  We deleted it. Line 23

3. Line 24-25, “The cross-resistance or multi-resistance spectrum and fitness cost caused by insecticide resistance are important” should be revised as “The understanding of cross-resistance or multi-resistance spectrum and fitness cost caused by an insecticide resistance is important”.

Author’s Response:  We have revised in the text.

Line 24-25: “The understanding of cross-resistance or multi-resistance spectrum and fitness cost caused by an insecticide resistance is important”

4. Line 27, “moderately resistant” should be revised as “moderate resistance”.

Author’s Response:  We have revised in the text.

Line 27, “moderate resistance”

5. Line 27, delete “with”

Author’s Response:  We have revised in the text.

Line 28, “accompanies with”

6. Line 35, what does an L is doing in this abbreviation. What does it stands for?. resistance ration is normally abbreviated as RR.

Author’s Response:  We revised RLR to RR in the full text.

7. Line 58, “organophosphorus” should be revised as “organophosphates”.

Author’s Response:  We have revised in the text.

Line 60, “organophosphates”

8. Line 61, “the” should be revised as “those”.

Author’s Response:  We have revised in the text.

Line 63, “those”

9. Line 110, “mother liquors” should be revised as “stock solutions”.

Author’s Response:  We have revised in the text.

Line 120, “stock solutions”

10. Line 157-165, I think don`t need repeat each RLR value that is already in the table, just mentions if there was a hight, medium, low or no resistance etc.

Author’s Response:  We have revised in the text. Line 166-174.

11. Line 235, “Earlier investigations of field-evolved insecticide resistance involved Ostrinia nubilalis” should be revised as “Earlier investigations of field-evolved insecticide resistance involved for example, Ostrinia nubilalis”.

Author’s Response:  We have revised in the text.

Line 250, “Earlier investigations of field-evolved insecticide resistance involved for example, Ostrinia nubilalis”.

12. Line 260, “We were unable to determine whether” should be revised as “It is unlikelly”.

Author’s Response:  We have revised in the text.

Line 280, “It is unlikelly”.

13. Line 263-265, The current sentence here suggests a multiple resistance rather than a cross-resistance, but if resistance to sulfoxaflor cannot be a result of selection by sulfoxaflor itself, then cross resistance sticks as the most likely explanation, specially if pyrethroids are used in wheat fields in China (likely are).

Author’s Response:  We have revised in the text.

Line 285-287, “On the contrary, the pyrethroids are widely and continuously used for control of wheat aphids in China [21], the resistance of field S. miscanthi population to sulfoxaflor are likely to be the result of selection by pyrethroids.”

14. Line 286, Authors may want to include a sentence, mentioned that while it need to be confirmed in further studies, the cross resistance and fitness reduction indicates metabolic resistance as more likely to be the type of resistance in this population.

Author’s Response:  We have revised in the text.

Line 308-310, “While it needs to be confirmed in further studies, the cross resistance and fitness re-duction indicates metabolic resistance as more likely to be the type of resistance in this population.”